# The Robust Italian Validation of the Coping Humor Scale (RI-CHS) for Adult Health Care Workers

**DOI:** 10.3390/ijerph19052522

**Published:** 2022-02-22

**Authors:** Roberto Burro, Alessandra Fermani, Ramona Bongelli, Ilaria Riccioni, Morena Muzi, Alessia Bertolazzi, Carla Canestrari

**Affiliations:** 1Department of Human Sciences, University of Verona, 37129 Verona, Italy; 2Department of Education, Cultural Heritage and Tourism, University of Macerata, 62100 Macerata, Italy; alessandra.fermani@unimc.it (A.F.); ilaria.riccioni@unimc.it (I.R.); morena.muzi@unimc.it (M.M.); carla.canestrari@unimc.it (C.C.); 3Department of Political Science, Communication and International Relations, University of Macerata, 62100 Macerata, Italy; ramona.bongelli@unimc.it (R.B.); alessia.bertolazzi@unimc.it (A.B.)

**Keywords:** stress, coping humor, scale validation, Rasch analysis, confirmatory factor analysis, invariance analysis

## Abstract

The Coping Humor Scale (CHS) is a seven-item tool widely used to assess the use of humor in coping with stressful situations. The beneficial effect of humor in buffering the impact of negative experiences has been investigated in several contexts and populations; for this reason, the CHS has been used in many languages, but its solid validation in Italian is still missing. Our study aimed at building a robust instrument to measure coping humor strategies among Italian health care workers, a category which has been particularly exposed to stressful situations in the last two years. The CHS translated into Italian was administered to a sample of 735 health care workers during the first wave of the COVID-19 pandemic in Italy. Confirmatory factor analysis and Rasch analysis were performed. As a result, a six-item Robust Italian Coping Humor Scale (RI-CHS) was validated and ready to use for future studies on Italian health care workers’ samples. This study gives evidence that our six-item solution works as a ruler (i.e., an instrument that meets the conditions of fundamental measurement in the context of the human sciences) to measure the degree to which Italian health care workers rely on humor to cope with stress.

## 1. Introduction

In the last few decades, there has been a growing interest in humor as a coping strategy against perceived stress and difficulties. According to the frame of positive psychology, adaptive humor has been found to be a positive mental health device in making the best out of a bad situation [1,2], and benevolent humor has been included in the values in action (VIA) classification of human strengths of character [3]. Humor as a strength of character lies in a competent playful attitude and a socially warm humorous style [4]. Along the course of the years, benign humor has been verified to play a significant role in coping with stressful events (e.g., [2,5,6]), contributing to resilience (e.g., [6,7]), predicting well-being and satisfaction (e.g., [8,9]), and affecting positively self-efficacy, which, in turn, elicits positive emotions [10]. In fact, approaching stressful situations, from daily hassles to traumatic experiences, with humor may positively impact emotion regulation, cognitive appraisal, and reappraisal of the demanding situation (e.g., [5,11,12,13,14]).

The beneficial effect of humor in buffering the impact of negative experiences has been explained on the basis of the cognitive processes involved in humor understanding. In fact, understanding a humorous stimulus implies changing its early interpretation so that a new and usually hidden meaning goes in the foreground and, as a result, a frame shifting [15,16] or a representational change [17] occurs. Likewise, a change in perspective on a problematic situation results when it is looked at through a humorous lens [18,19]. The same process applies to a problem-solving activity, which matches up humor understanding in many respects [20,21,22]. Therefore, it is not surprising that people who report using coping humor also report they feel they have solved the problem [23], besides taking control over the problematic situation, as a result of the coping power of humor [6]. Furthermore, reframing a negative situation via humor elicits a cognitive distraction from a negative mood, which, in turn, attenuates negative emotions [24]. In addition, negative emotions are regulated through humor since it positively influences social relationships [7,23,25].

The growing interest in studying the beneficial effects of positive humor and its coping power goes hand in hand with its assessment as a coping strategy. Back in 1983, the Coping Humor Scale (CHS) was the first tool validated to measure the use of humor as a coping mechanism [13]. Later, further scales were validated to specifically measure the use of humor as a coping strategy. For example, the two-item humorous coping scale is a subscale of the brief COPE questionnaire, which measures more general aspects of coping [26]; the Waterloo Uses of Humor inventory is a 21-item scale developed by Thomas [27], to assess three facets of coping humor, namely perspective change, aggressive, and avoidant coping humor. Other measures were developed to assess coping humor in specific contexts, such as intimate relationships (two subscales of the relational humor inventory, developed by De Koning and Weiss, [28]) and the work environment (25-item questionnaire of occupational humorous coping, developed by Doosje, De Goede, Van Doornen, and Goldstein [29]). However, none of them were widely used and validated in so many languages as the CHS (for an overview, see [30]).

The CHS is a seven-item self-report questionnaire aimed at measuring the degree to which individuals report using humor to cope with stress. It is a reliable and valid measure and contains an adequate number of items so that the questionnaire is not too long nor too short to fill in. Its internal consistency (Cronbach’s alpha) ranges from 0.60 to 0.70 [30]. However, further empirical studies have highlighted that leaving out Item 4, which was shown to be interpreted in inconsistent ways, improves internal consistency [30]. For example, a confirmatory factor analysis carried out on the remaining six items showed adequate construct validity [31]. Moreover, the test–retest (Pearson correlation) reliability of CHS, measured over 12 weeks, is 0.80 [32]. Construct validity for CHS is considerable, and it includes positive correlations with external perception of an individual’s use of coping humor, positive associations with other coping styles, and negative associations with neuroticism (for an overview, see [29,33].

Although the scale has been translated into Italian by several authors and has been used in numerous studies to assess humorous coping skills in response to cataclysmic stressors (earthquakes, COVID-19 pandemic), e.g., [34,35,36], psychological distress [37,38], and life stages [39], a solid Italian validation of the CHS is still missing and necessary.

The present study aims to fill in this gap by gathering a robust Italian validation of CHS for health care workers (HCWs). Medical setting is the focus of the present study, as also in workplaces, humor resulted to be an efficacy strategy to buffer job stress, to enhance job satisfaction, and therefore, to contribute to workplace well-being (e.g., [40,41,42,43]). Likewise, the use of coping humor—in its diverse facets—by HCWs has been found to promote subjective well-being [23,35,44], to promote beneficial interactions and rapports with patients, when properly negotiated [23,44,45,46,47]), and to enhance social cohesion among co-workers [23,44], even via gallows humor [48].

HCWs belong to a category of workers exposed to psychological stress, in particular after epidemic or pandemic outbreaks [49]. Consistent with the results of considerable research on the psychological outcomes of past pandemics among healthcare workers, e.g., [50,51], numerous studies carried out during the COVID-19 pandemic have revealed similar negative effects (e.g., stress, anxiety, depression, distress, insomnia, emotional exhaustion, burnout, PTSD e.g., [49,52,53,54,55,56,57,58,59] among healthcare workers; specifically, among those employed on the front lines and in areas most affected by the virus. Given the massive studies carried out on this category of workers after the COVID-19 outbreak, aiming at determining their coping strategies against the psychological impact of COVID-19-related difficulties, an Italian CHS for HCWs is required. Despite the strong psychological impact of COVID-19 outbreak among Italian HCWs [55,60], up to now, only one study has been carried out on how Italian HCWs use coping humor [35]. Therefore, future studies on the use of coping humor in Italian medical settings are expected, and they will benefit from an Italian CHS for HCWs.

### 1.1. Measurement of Coping Humor Strategy

The main purpose of the following work is to develop an Italian validation of CHS for HCWs. Precisely, our intention is to build a robust instrument to measure coping humor strategy (i.e., an instrument that satisfies the conditions of fundamental measurement) [61]. For this reason, we called our scale, Robust Italian Coping Humor Scale (RI-CHS). According to these conditions, a measurement should not be derived from other measurements and should be obtained by an additive measurement operation [62]. Fundamental measurement is usual in the physical and natural sciences, while it is often snubbed in the bio-psycho-social sciences. In order to achieve this goal, we used the Rasch model framework [63,64] after performing a confirmatory factor analysis (CFA).

In the literature, this procedure is well known (e.g., [65,66,67,68,69]).

Among the advantages of this procedure, there is the possibility of obtaining sample-free and test-free measures on an interval logit scale [70]). Since we were interested in the Italian validation of a well-known one-dimensional scale, and in order to fully exploit the sample, we avoided performing a previous exploratory factor analysis (EFA) that would not have brought any concrete advantage.

To apply a Rasch analysis, there are a series of assumptions that must be verified precisely: the presence of monotonicity [71,72], the local independence [73,74,75], the unidimensionality [76,77], and the absence of differential item functioning (DIF) [78,79]. If one or more of the assumptions are not satisfied, there is the possibility to intervene using a set of modification strategies [80,81]), to adjust the violations of monotonicity (e.g., item rescoring) of local independence (e.g., item grouping or “testlets” creation) and the presence of DIF (e.g., item splitting).

Later, the analysis of the standardized residuals for the responses associated with individual items across persons is useful to show patterns in unexpected and expected responses [82]. It is then possible to study the items’ performance analyzing the infit-MSQ index (i.e., mean square inlier-sensitive fit) and the outfit-MSQ index (i.e., mean square outlier-sensitive fit) [83].

As a last resort, when previous operations have not been effective or have shown negative results, it is possible to delete critical items, and repeat the above procedure iteratively. In addition, the person separation index (PSI) [84,85] shall be used to evaluate the reliability of the instrument. If all assumptions are met, the model-data fit can be evaluated using Andersen’s [86] likelihood ratio test. Finally, it is possible to transform the raw scores into an interval logit scale [87] that satisfies the conditions of fundamental measurement.

## 2. Materials and Methods

### 2.1. Participants/Data Collection

Data were collected in compliance with the principles of the Declaration of Helsinki (https://www.wma.net/what-we-do/medical-ethics/declaration-of-helsinki/, consulted on 14 January 2022), European and Italian privacy legislation (i.e., EU Reg. 679/2016, GDPRD, and Legislative Decree No. 196/2003, Personal Data Protection Code), and the APA Code of Ethics. The research has been approved by a PhD curriculum meeting in psychology, communication, and social sciences (University of Macerata. Protocol code n. 19435, 3 August 2020).

On 15 May 2020, Italian nurses and physicians (enrolled in professional associations and orders) were invited to participate in an online survey that was proposed using LimeSurvey software (version 3.22; LimeSurvey GmbH, 2012 [88]) on a LAMP (Linux, Apache, MySQL, PHP) web-server. The HTTPS protocol and secure sockets layer (SSL) were adopted. Data collection ended on 30 June 2020.

Inclusion criteria: physician and nurse members of the Italian professional orders and of the principal associations for physicians and nurses (e.g., Italian Society of Anesthesia, Analgesia, Resuscitation and Intensive Care), working in Italian hospitals, nursing and retirement homes, clinics in the territory etc., during the first Italian lockdown for COVID-19.

Exclusion criteria: physicians and nurses not enrolled in Italian professional orders; physicians and nurses not on duty during the first Italian lockdown for COVID-19; other healthcare workers different from physicians and nurses (e.g., psychologists, socio-health workers, secretarial staff, etc.).

The chain-referral sampling method was used. The first questions collected socio-demographic and employment information. Later, the CHS was administered. We adapted the original CHS scale into the Italian language using a forward and backward translation process to guarantee correspondence between Italian and English original versions. All the items of the questionnaire were compulsory. Each item was rated on a four-point scale (1 = strongly disagree; 2 = mildly disagree; 3 = mildly agree; 4 = strongly agree).

There were no incentives for compiling the survey. Before filling in the questionnaire, the respondents had to agree to informed consent. The estimated average time for compiling the questionnaire was approximately 5 min.

### 2.2. Sample Characteristics

A total of 735 participants compiled the questionnaire, 516 (70.2%) were women and 219 (29.8%) men. The mean age was 45.39 (ranging from 21 to 81, SD = 12.04). The majority of the participants were married (45.1%), had children (57.9%), and declared to be believers and occasional practitioners (38.4%). Furthermore, 75.9% worked as a nurse, 70.2% in hospitals and care services in the northern region of Italy, and 51.8% in the area of medical specialties, with 47.6% working for more than 20 years. Moreover, 41.1% of them claimed to have worked in a COVID-19-dedicated ward (i.e., on the frontline), while 58.9% were in other wards (see Table 1 for more details). As the questions were mandatory to complete the questionnaire; there were no missing data. The data were collected during the first wave of the COVID-19 pandemic in Italy in 2020.

## 3. Results

### 3.1. Data Analysis

#### 3.1.1. Confirmatory Factor Analysis

All the analyses were carried out using R software, Version 4.1.2 (R Core Team, 2021 [89]).

Through preliminary analyses, we verified that our data were suitable for factor analyses: Bartlett’s test of sphericity, χ^2^(21) = 1608.12, *p* < 0.001; Kaiser–Meyer–Olkin, KMO = 0.81. Bartlett’s test of sphericity verifies whether the correlation matrix is an identity matrix, which indicates that the factor model is inappropriate. The KMO measure of sampling adequacy tests whether the partial correlations among the variables are small.

To test the internal factor structure of the instrument, we ran confirmatory factor analysis (CFA) on all seven items of the scale (see columns 2 and 4 of Table 5).

The CFA was carried out using the Diagonally Weighted Least Squares estimator (DWLS, a method of estimating the parameters specifically designed for ordinal data). We tested the adequacy of confirmatory solutions by means of the following different fit-indexes: the Root-Mean-Square Error of Approximation (RMSEA, that assesses how much a hypothesized model differs from a perfect model), the Comparative Fit Index (CFI, that analyzes the model fit by examining the discrepancy between the data and the hypothesized model), the Tucker-Lewis Index (TLI, that analyzes the discrepancy between the chi-squared value of the hypothesized model and the chi-squared value of the null model), the Standardized Root Mean Residual (SRMR, that represents the square root of the discrepancy between the sample covariance matrix and the model covariance matrix using standardized values), the Goodness of Fit Index (GFI, that measures the fit between the hypothesized model and the observed covariance matrix), and the Adjusted Goodness of Fit Index (AGFI, that is a correction of GFI, which is conditioned by the numerosity of indicators of each latent variable). The threshold values to assess the goodness of fit were: ≤0.08 for RMSEA and SRMR, ≥0.95 for TLI and GFI, ≥0.90 for CFI and AGFI [90,91,92,93]. As recommended by the literature, we proceeded by considering only those models showing fit indices below the cut-off values for RMSEA and SRMR and above the cut-offs for CFI, TLI, GFI and AGFI as good.

According to Kline [94], for proper CFA, the minimum ratio between the number of observations and the number of parameters should be preferably 10:1. In our case, we had a ratio of 26.25 (735 observations and 28 model parameters). For this reason, the size of our sample was appropriate.

Four of the six fit-indexes considered were good: CFI = 0.970, TLI = 0.956, GFI = 0.979, and AGFI = 0.939. The remaining two were inadequate: RMSEA = 0.137 and SRMR = 0.097. Taking into account the standardized factor loadings of each of the seven items, we saw that Item 4 did not perform adequately (z-value = 0.834, *p* = 0.404, standardized factor loadings = 0.018, see Figure 1a), consistent with previous studies [30,31,95]).

Because of this, we decided to eliminate Item 4, and we recalculated the model using a six-item solution. In this case, all fit-indexes were good: CFI = 0.994, TLI = 0.990, GFI = 0.995, AGFI = 0.982, RMSEA = 0.076, and SRMR = 0.047. All six items performed well (see Figure 1b).

Using RMSEA as effect size and alpha = 0.05, the results of the post-hoc power analysis show that a sample size of N = 735 is associated with a power larger than 99.99%.

#### 3.1.2. Measurement Invariance Analysis

To verify the measurement invariance (MI) of the instrument with respect to gender, that is to assess whether the six-item solution model was invariant and generalizable across males and females, a multigroup confirmatory factor analysis (MG-CFA) was performed, which began with a separate baseline model for each group. The configural invariance model was established when the same factorial pattern was specified for each group but with factor loadings and intercepts free across samples; in the metric invariance model, factor loadings were constrained to be equal across groups; in the scalar invariance model, factor loadings and intercepts were constrained to be equal across conditions.

To compare the three models, we considered the differences between three indices, CFI, SRMR, and RMSEA (see Table 2). The MI is verified if there exists a difference in CFI less or equal to 0.010, a difference in RMSEA less or equal to 0.015, and a difference in SRMR less or equal to 0.030 for testing metric invariance and less or equal to 0.010 for testing scalar invariance [96].

The results in Table 2 show that the six-item solution was invariant and generalizable across males and females.

#### 3.1.3. Internal Consistency Reliability

We estimated internal consistency reliability of the six-item solution by the Cronbach’s alpha and McDonald’s omega indexes. The threshold values to assess the indexes were as follows: >0.90 excellent; < 80–90 > good; < 70–80 > acceptable; < 60–70 > questionable; <0.60 bad. The Cronbach’s alpha and the McDonald’s omega reliability indexes were equal to 0.801 and 0.811, respectively. This suggests that the six-item solution was good in terms of reliability.

#### 3.1.4. Rasch Analysis

In order to make the previous instrument more robust (i.e., to obtain an instrument that satisfies the conditions of fundamental measurement), a Rasch analysis (RA) was conducted using the partial credit model [87].

RA is the process of testing statistically whether the data fit the assumptions and requirements of a mathematical model named after its developer, the Danish mathematician Georg Rasch [64].

##### Checking Requirements of the 6-Items Solution

First, we checked the monotonicity looking at whether the thresholds (i.e., the transition points between two different scores in the response scale) were correctly ordered. To do this, we used the person–item map (see Figure 2).

The person–item map indicated that the scores of the Item 1 had non-ordered thresholds (Figure 2a). Thus, we rescored Item 1 because it violated the monotonicity assumptions. Starting from Figure 2a, two types of rescoring were possible for Item 1: first, the response scale changed from 1, 2, 3, 4 to 1, 1, 2, 3 (i.e., 2 becomes 1, 3 becomes 2, and 4 becomes 3); second, the response scale changed from 1, 2, 3, 4 to 1, 2, 2, 3 (i.e., 3 becomes 2 and 4 becomes 3). These operations allowed us to satisfy the monotonicity requirement in two different ways (Figure 2b,c). The second solution has been preferred because the two thresholds of Item 1 were more distant from each other in Figure 2c than in Figure 2b.

In a second step, we then examined the local independence (i.e., the items in a scale should not be related to each other) by analyzing the correlations between the items’ residuals [73,74,75]). Since correlations were never larger than 0.30 (−0.37 < r < 0.02), there was evidence of local independence.

The third step was to verify the unidimensionality of the six-item solution after the rescoring operation, using a principal components analysis (PCA) of the correlations among standardized residuals from Rasch model analyses. PCA of residuals evaluates the degree to which additional dimensions may have contributed to item responses. In the context of Rasch analyses, PCA of residuals describe the eigenvalues as contrasts, because they reflect contrasting patterns of responses to the principal latent variable [77,97,98,99]. Unidimensionality occurs when all the contrasts have a value less than 2 [100]. In our case, the largest contrast was 1.52. This result provides evidence of unidimensionality.

In the fourth step, the absence of differential item functioning (DIF) was checked, i.e., we tested whether the instrument measured different subgroups of participants in the same way. In this regard, we calculated the Standardized P-DIF statistic for gender (−0.079 < St-P-DIF < 0.095). Given that for all items the index fell between −0.10 and 0.10 [101], the instrument functioned similarly for males and females.

The fifth step checked the standardized residuals for the responses given by persons to each item. A separate plot was produced for each item (see Figure 3). Standardized residuals with zero values indicate that the observed responses coincided with the model-expected responses. When the values exceed ±2, they were interpreted as indicating statistically significant unexpected responses. For all items, the vast majority of standardized residuals did not exceed the ±2 range. The total percentage of misfitting persons was equal to 4.454.

On the sixth step, we studied the items’ performance by analyzing the indices infit-MSQ and outfit-MSQ. When these indices are higher than 1.60, it means that the items underfit the Rasch model (i.e., the data are less predictable than the model expects); when they are lower than 0.40, it means that the items overfit the Rasch model (i.e., the data are more predictable than the model expects) [83]. If an item shows underfit/overfit, it is advisable to remove it. Table 3 shows that all our items had indices that fell within the appropriate range.

Concerning reliability, the PSI was adequate (0.77).

After all these checks, we tested the fit of the data to the Rasch model using Andersen’s likelihood ratio test (χ^2^(16) = 21.592, *p* = 0.157) [86]. Since the value of χ^2^ is not significant, we can conclude that the data-model fit is good.

Finally, we transformed the raw scores of the six-item solution into an interval logit scale that satisfies the conditions of fundamental measurement [87], as shown in Table 4. For convenience, the logit scores were scaled from 1 to 10 (considering that Item 1 was rescored and reversed). For example, if a participant gets a total raw score of 10, its value should be replaced with 3.682 logits; a total raw score of 15 corresponds to 5.300 logits, and so on.

#### 3.1.5. Criterion Validity

In order to assess the criterion validity, we calculated the intercorrelations between the RI-CHS (logit score) and the two-coping humor items (item 18 and item 28) of the brief COPE questionnaire [26]. Both correlations are positive and significant (Pearson’s r = 0.438, *p* < 0.001 between RI-CHS score and Item 18; Pearson’s r = 0.435, *p* < 0.001 between RI-CHS score and Item 28).

At this point, it is possible to use the RI-CHS as a ruler (i.e., an instrument that meets the conditions of fundamental measurement in the context of the human sciences) to measure the degree to which Italian health care workers rely on humor to cope with stress (see columns 4 and 5 of Table 5).

## 4. Discussion

Our study provides a six-item robust instrument to measure coping humor strategies among Italian HCWs, by adapting and validating the seven-item original version of the CHS by Martin and Lefcourt [13]. The analysis we conducted revealed that Item 4 of the original scale is not adequate to measure HCWs’ perception that more sense of humor would make their life easier (Item 4: “I must admit my life would probably be a lot easier if I had more of a sense of humor”), and therefore, it does not contribute to the understanding of the coping humor strategies. Similarly, Martin [30], who reviewed a set of studies conducted mostly on samples of university students, found that the internal consistency of the CHS (alpha ranging from 0.60 to 0.70) increased without Item 4. The same was found by Chen and Martin (2007) [95], who suggested omitting Item 4. Nezler and Derks [31], who evaluated the psychometric properties of the CHS, confirmed that Item 4 was inadequate to measure the latent construct of the scale and found a good internal consistency of the six-item scale (omitting Item 4), with a Cronbach’s alpha of 0.75. Finally, the inadequateness of Item 4 has been confirmed also in a sample of 625 HCWs [35].

Only recently, coping humor among Italian HCWs has been investigated: in a study, the Italian HCWs who reported higher use of humor-based coping strategies perceived the situation as less stressful in comparison with those who reported less use of coping humor, during the COVID-19 outbreak [35]. The general protective power of humor against life-threatening and stressful situations is an acknowledged result within the literature, and it has also been confirmed in studies on HCWs (e.g., [23,44,48]). As a result, humor-based interventions to promote and support mental health and well-being have been put forward (for an overview, see [102]) also for HCWs (e.g., [103]), and their efficacy investigated (e.g., [104,105]). In order to plan effective humor-based interventions for HCWs, the RI-CHS is a useful tool to measure the starting coping humor level of the HCWs to whom humor-based interventions are targeted. Moreover, in view of the importance of coping humor and its limited studies among Italian HCWs, further studies in need of the RI-CHS as a valid instrument for measuring humor coping are expected.

RI-CHS enjoys two prominent qualities: it is sample-free (the item-difficulty estimates are independent of the specific sample of persons used in the study) and test-free (the person-ability estimates are independent of the particular sample of items used in the study). The actual distribution of the items and persons is irrelevant. For this reason, we believe it can also be used in non-health care contexts.

The six items of RI-CHS do not refer specifically to the health care setting after all. Moreover, similar psychometric properties have been found with the original seven-item scale (i.e., the inadequateness of Item 4), which has been validated outside the health care setting. Therefore, it can be argued that RI-CHS is valid for Italian populations other than HCWs. However, this assumption needs further verification.

It should also be considered that having validated the scale on a sample so exposed to strain and adversities, due to the pandemic, probably amplifies the scale’s ability to measure the use of humor as a coping strategy among Italian HCWs. In fact, the RI-CHS is a self-reported measure, and reporting one’s ability to use humor to cope against stress and difficulties when they are particularly present makes the detection of such ability more authentic. For this reason, the survey was carried out during a pandemic.

One real limitation of the RI-CHS may be that it consists of only six items. This allows only one dimension of coping humor to be captured and prevents effective exploration of multidimensional contexts. For example, the communicative dimension of coping humor is poorly represented: only item 3 (“I usually look for something comical to say when I am in tense situations”) refers to the communicative aspect, but it does not entail the type of humor strategies nor to the type of relationship between interactants. Identifying the types of humorous coping strategies used by HCWs with colleagues or patients, that are perceived as most effective in reducing work-related stress, is an important further step in broadening our understanding of multifaceted aspects of coping humor at work. Moreover, the point of view of patients is an important dimension to take into account when coping humor is used by HCWs with them, in order to promote humorous coping strategies that are also helpful for the well-being of patients. Further studies could investigate these aspects in order to get a clearer qualitative picture of effective humor coping strategies after detecting, through the RI-CHS, how frequently humor coping is used among HCWs.

## 5. Conclusions

Our study enabled us to assess the ability to use humor as a coping strategy among HCWs, by adapting into Italian and validating the seven-item CHS. Taking advantage of the strengths of the Rasch model, the RI-CHS, a robust instrument made of six items for Italian HCWs, was put forward. Our study adds to the existing literature on coping humor by filling in a gap related to the missing validation of the CHS into Italian. From the theoretical point of view, the study presented here confirmed the inadequateness of an item out of seven, as pointed out in some relevant studies, and extends previous literature by providing a robust six-item tool. From an applied perspective, the RI-CHS has a lot of potential, as it can be useful for developing interventions to support HCWs’ ability to use humor as a coping strategy, as well as for further studies interested in assessing Italian HCWs’ coping humor.

## Figures and Tables

**Figure 1 ijerph-19-02522-f001:**
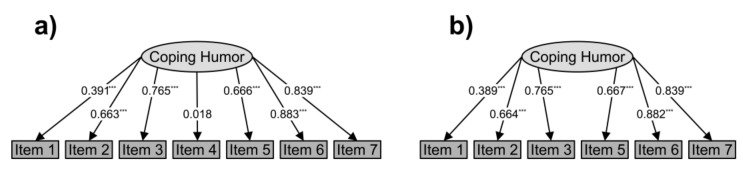
(**a**) Factorial model of the seven-item solution. (**b**) Factorial model of the six-item solution. The digits represent standardized factor loadings. *** *p* < 0.001.

**Figure 2 ijerph-19-02522-f002:**
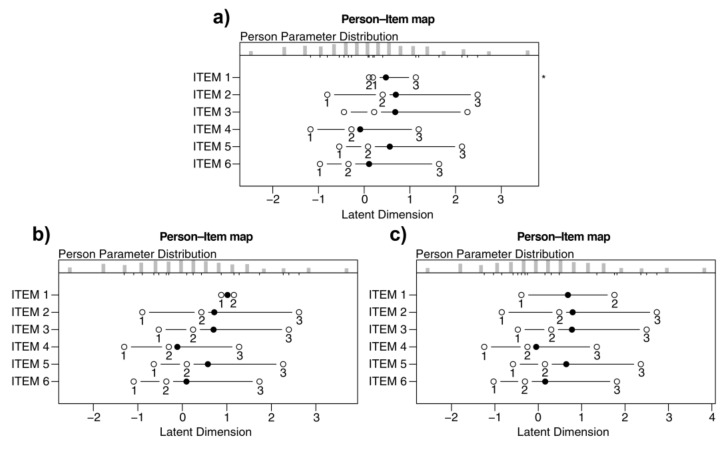
(**a**) Person–item map relating to the six-item solution. (**b**) Person–item map relating to the six-item solution after the first rescoring. (**c**) Person–item map relating to the six-item solution after the second rescoring. The solid circles represent the locations of the items’ discriminatory capacities. The open circles represent the thresholds. The asterisk indicates a problematic item with non-ordered thresholds.

**Figure 3 ijerph-19-02522-f003:**
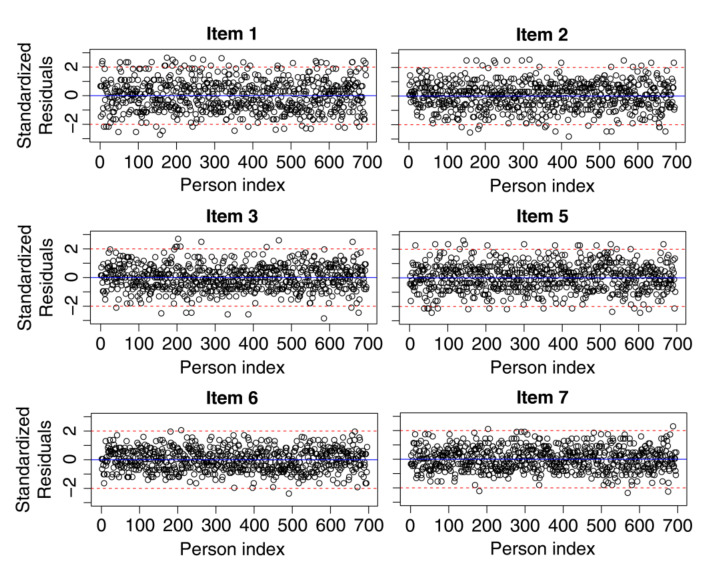
Plot of the standardized residuals. The *y*-axis shows values of the standardized residuals for each item, and the *x*-axis shows the persons ordered by their identification number.

**Table 1 ijerph-19-02522-t001:** Descriptive statistics of the sample characteristics.

Variables	n (%)
**Total**	735 (100%)
**Socio-demographic characteristics**	
*Gender*	
Female	516 (70.2%)
Male	219 (29.8%)
*Age*	
18–30	137 (18.6%)
31–40	141 (19.2%)
41–50	196 (26.7%)
51–60	208 (28.3%)
>60	53 (7.2%)
*Marital status*	
Married	332 (45.1%)
Unmarried	202 (27.5%)
Domestic partner	115 (15.7%)
Divorced/separated	72 (9.8%)
Widower/widow	14 (1.9%)
*Children*	
Yes	426 (57.9%)
No	309 (42.1%)
*Religion*	
Believer occasionally practitioner	282 (38.4%)
Believer non-practitioner	175 (23.8%)
Non-Believer	122 (16.6%)
Believer practitioner	116 (15.8%)
Prefer not to answer	40 (5.4%)
**Job characteristics**	
*Place of work*	
North Italy	516 (70.2%)
Centre Italy	138 (18.8%)
South Italy	81 (11.0%)
*Job position*	
Nurse	558 (75.9%)
Physician	177 (24.1%)
*Job area*	
Medical specialties	381 (51.8%)
Diagnostic and therapeutic specialties	155 (21.1%)
Surgical specialties	114 (15.5%)
Primary care nurse serv.	85 (11.6%)
*Seniority*	
More than 20 years	350 (47.6%)
Less than 5 years	162 (22.0%)
10–20 years	131 (17.8%)
5–10 years	92 (12.6%)
**Job exposure to COVID-19**	
*Wards*	
Worked in COVID-19-dedicated wards	302 (41.1%)
Worked in other wards	433 (58.9%)

**Table 2 ijerph-19-02522-t002:** Results of Measurement Invariance analyses across gender (males, females). ΔCFI, ΔRMSEA and ΔSRMR = differences in Comparative Fit Index (CFI), Root-Mean-Square Error of Approximation (RMSEA), and Standardized Root Mean Residual (SRMR).

Models	ΔCFI	ΔRMSEA	ΔSRMR
Configural	−0.001	0.009	0.006
Metric	−0.000	−0.006	0.004
Scalar	−0.002	−0.005	−0.003

**Table 3 ijerph-19-02522-t003:** Infit-MSQ and outfit-MSQ of each item.

Original CHS Item Number	Infit-MSQ	Outfit-MSQ
1 (reverse)	1.356	1.399
2	0.949	0.959
3	0.798	0.788
5	0.946	0.941
6	0.608	0.605
7	0.650	0.642

**Table 4 ijerph-19-02522-t004:** Conversion table of total raw scores to Robust Italian Validation of the Coping Humor Scale (RI-CHS) interval scale logit scores.

Total Raw Scores	Interval Logit Scores
6	1
7	1.909
8	2.743
9	3.271
10	3.682
11	4.035
12	4.359
13	4.670
14	4.981
15	5.300
16	5.637
17	6.001
18	6.400
19	6.848
20	7.368
21	8.013
22	8.969
23	10

**Table 5 ijerph-19-02522-t005:** Original CHS items and their response scale (columns 2 and 3) vs. RI-CHS items and their response scale (columns 4 and 5). Note: the RI-CHS Item 1 was a reverse-item, and it was rescored.

Items Number	CHS Items (English)	CHS Response Scale	RI-CHS Items (Italian)	RI-CHS Response Scale
1 (reverse item)	I often lose my sense of humor when I am having problems	4–Strongly disagree 3–Mildly disagree 2–Mildly agree 1–Strongly agree	Perdo il senso dell’umorismo quando ho problemi	3–Molto in disaccordo 2–Un poco in disaccordo 2–Un poco d’accordo 1–Molto d’accordo
2	I have often found that my problems have been greatly reduced when I try to find something funny in them	1–Strongly disagree 2–Mildly disagree 3–Mildly agree 4–Strongly agree	Ho riscontrato che i miei problemi si sono fortemente ridotti quando ho provato a trovare in essi qualcosa di divertente	1–Molto in disaccordo 2–Un poco in disaccordo 3–Un poco d’accordo 4–Molto d’accordo
3	I usually look for something comical to say when I am in tense situations	1–Strongly disagree 2–Mildly disagree 3–Mildly agree 4–Strongly agree	Cerco qualcosa di comico da dire quando sono in situazioni tese	1–Molto in disaccordo 2–Un poco in disaccordo 3–Un poco d’accordo 4–Molto d’accordo
4 (reverse item)	I must admit my life would probably be a lot easier if I had more of a sense of humor	4–Strongly disagree 3–Mildly disagree 2–Mildly agree 1–Strongly agree	Devo ammettere che la mia vita sarebbe più facile se avessi maggiore senso dell’umorismo	NO RESPONSE SCALE This item has been excluded from RI-CHS
5	I have often felt that if I am in a situation where I have to either cry of laugh, it’s better to laugh	1–Strongly disagree 2–Mildly disagree 3–Mildly agree 4–Strongly agree	Mi è capitato di pensare che, se sono in una situazione dove si può piangere o ridere, è meglio ridere	1–Molto in disaccordo 2–Un poco in disaccordo 3–Un poco d’accordo 4–Molto d’accordo
6	I can usually find something to laugh or joke about even in trying situations	1–Strongly disagree 2–Mildly disagree 3–Mildly agree 4–Strongly agree	Riesco a trovare qualcosa su cui ridere o scherzare persino in situazioni difficili	1–Molto in disaccordo 2–Un poco in disaccordo 3–Un poco d’accordo 4–Molto d’accordo
7	It has been my experience that humor is often a very effective way of coping with problems	1–Strongly disagree 2–Mildly disagree 3–Mildly agree 4–Strongly agree	Fa parte della mia esperienza pensare che l’umorismo sia spesso una via efficace per fronteggiare i problemi	1–Molto in disaccordo 2–Un poco in disaccordo 3–Un poco d’accordo 4–Molto d’accordo

## Data Availability

Not applicable.

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
