# Peer review of "The Robust Italian Validation of the Coping Humor Scale (RI-CHS) for Adult Health Care Workers"

_ijerph, 2022, doi:10.3390/ijerph19052522_

Round 1

Reviewer 1 Report

This was a fine study to review. I only have some minor issues that could make the manuscript better:

  • When discussing reliability in the literature part, please make clear if it's e.g. Cronbach's Alpha or another measure for reliability.
  • Freud didn't really use 'coping' mechanisms -> this is rather a concept used by Erikson. Freud's original Abwehrmechanismen is often translated as a defence mechanism and has a negative connotation, while Erikson's coping mechanisms can have both a positive and negative connotation.
    Do check https://annals-general-psychiatry.biomedcentral.com/articles/10.1186/s12991-020-00307-1#:~:text=Traditionally%2C%20defense%20mechanisms%20are%20patterns,based%20on%20cognition%20%5B7%5D. on this theme.  

Author Response

Response to Reviewer 1 Comments
Thank you for your valuable contribution that improves this manuscript.

Point 1: When discussing reliability in the literature part, please make clear if it's e.g. Cronbach's Alpha or another measure for reliability.

Response 1: Done, added in the resubmitted manuscript (highlighted in yellow, line 73 and line 77).

Point 2: Freud didn't really use 'coping' mechanisms -> this is rather a concept used by Erikson. Freud's original Abwehrmechanismen is often translated as a defence mechanism and has a negative connotation, while Erikson's coping mechanisms can have both a positive and negative connotation.
Do checkhttps://annals-general-psychiatry.biomedcentral.com/articles/10.1186/s12991-020-00307-1#:~:text=Traditionally%2C%20defense%20mechanisms%20are%20patterns,based%20on%20cognition%20%5B7%5D on this theme. 

Response 2: We agree with the reviewer that the connections between coping mechanism and defence mechanism is controversial. Since this aspect is entirely peripheral in the paper, we preferred to delete the reference to Freud and defence mechanisms (highlighted in yellow and crossed out, line 31 in the resubmitted manuscript).

Reviewer 2 Report

The manuscript titled “The Robust Italian validation of the Coping Humor Scale (RI-2 CHS) for adult health care workers” aiming to carry out a validation study for the Coping Humor Scale (RI-2 CHS) for application in care workers.

Author(s) administered the scale to a sample of 735 health care workers during the first wave of the COVID-19 pandemic in Italy and provided some psychometric properties.

I carefully read the manuscript and I suppose it may be of interest for readers of International Journal of Environmental Research and Public Health. Even so, before publishing it as a research article it could be worth considering some points. Below there are my comment and suggestion.

Introduction

Lines 81-83: It may be useful to further explain in which Italian context the English version has been used

Lines 92-98: Further details on effects of the pandemic on HCW is needed. Please, provide an overview of the situation related to the mental health in those job categories.

Participants//Data collection

Inclusion and exclusion criteria are missing. Please add them.

Sample characteristics

The reason for investigating for religion might be somewhat explained. A mention in introduction also may be useful to be added.

Results

Analyses considering demographics may be appropriate. Otherwise, it is not clear why that information has been collected.

Confirmatory Factor Analysis

  • Why an exploratory factor analysis hasn’t been performed?
  • Also, I suggest briefly explaining the meanings and cut-offs of the fit-indices (lines 180-186)
  • The model tested was a single-factor model. Please explain the reason on the basis on this choice since EFA is missing. May be for convenience or previous literature-based? Please specify.

Discussion

The reason why a validation on HCW (specific sample) during pandemic (very specific period) is missing. Please elaborate more this part.

Author Response

Response to Reviewer 2 Comments
Thank you for your valuable contribution that improves this manuscript.

Point 1: Introduction. Lines 81-83: It may be useful to further explain in which Italian context the English version has been used.

Response 1: We explained further by adding some text in the resubmitted manuscript (highlighted in yellow, lines 82-86)

Point 2: Introduction. Lines 92-98: Further details on effects of the pandemic on HCW is needed. Please, provide an overview of the situation related to the mental health in those job categories.

Response 2: We explained further by adding some text in the resubmitted manuscript (highlighted in yellow, lines 96-101)

Point 3: Participants/Data collection. Inclusion and exclusion criteria are missing. Please add them.

Response 3: Done, we have added "inclusion criteria" and "exclusion criteria" in the resubmitted manuscript (highlighted in yellow, lines 158-166)

Point 4: Sample characteristics. The reason for investigating for religion might be somewhat explained. A mention in introduction also may be useful to be added.

Response 4: Among socio-demographic items, we included not only questions regarding marital status and having or not having children, but also a question about participants’ religiosity. These variables may indeed not only have played a role in determining certain psychological outcomes among HCWs, but they may also have played an important role in the ways in which HCWs dealt with the highly stressful situation. With respect to religious beliefs, the brief-cope scale - which we are preparing to validate in Italian- contains two items specifically devoted to testing this coping dimension.

Point 5: Results. Analyses considering demographics may be appropriate. Otherwise, it is not clear why that information has been collected.

Response 5: We reported the data in Table 1 to give an accurate idea of the sample structure. We believe that the categories reported are important to the generalizability of the sample. We plan to analyze these data (in detail) in another article and not "skipping steps" in this article. For example, some categories need more data for the purpose of adequate power-analysis. Instead, we believe that the emphasis of this article should be on the prominent measurement qualities of the scale. To do things right we need several analyses and this paper already reports several. There is the risk of transfiguring the article meaning.

Point 6: Confirmatory Factor Analysis. Why an exploratory factor analysis hasn’t been performed?

Response 6: Please see response 8 below.

Point 7: Confirmatory Factor Analysis. Also, I suggest briefly explaining the meanings and cut-offs of the fit-indices (lines 180-186)

Response 7: We explained further by adding some text in the resubmitted manuscript (highlighted in yellow, lines 202-214 and lines 216-218)

Point 8: Confirmatory Factor Analysis. The model tested was a single-factor model. Please explain the reason on the basis on this choice since EFA is missing. May be for convenience or previous literature-based? Please specify.

Response 8: We did not perform an Exploratory Factor Analysis (EFA), because we would have got no advantage. In this regard, it should be considered that:
- we have validated (in Italian) a scale that previous studies have shown to be a single factor scale;
- a rule of thumb says that a factor should have at least 3 items. Therefore, in our case there is the possibility of two factors at most, but we have no (in the literature and in our data) evidence of this. Having 2 factors with 3 or 4 items means, however, having 2 factors that risk do not perform very well;
- doing an EFA and then a CFA would have meant dividing the sample into 2 halves (one for the EFA and one for the CFA) with the risk of reducing the generalizability of the results.
In this regard, we have added text in the resubmitted manuscript (highlighted in yellow, lines 121-124)

Point 9: Discussion. The reason why a validation on HCW (specific sample) during pandemic (very specific period) is missing. Please elaborate more this part.

Response 9: At the end of the Discussion, we stressed that having carried out the survey during a pandemic could have amplified the scale’s ability to measure the use of humor as a coping strategy. All the additions (in the resubmitted manuscript) are highlighted in yellow (lines 388-393 and lines 396-408).

Reviewer 3 Report

Dear authors! The topic and the goals of the study are relevant and important, especially in conditions of world pandemic. I only have very little suggestions to improve your work. I think it is necessary to include more theoretic positions about the importance on the sense of humor in professional work. Also, it is important to present clearly each question or item of the test for the readers. I suggest to start the presentation of the data with general qualitative features of the responses to each item. I suggest to include concrete comments about the usefulness of each item or a kind of critical reflection about the items.  Probably, proposals about future studies or inclusion of new items or new design of the tests. It is also useful to think that psychological study may not be based only on self reports, but also a confirmation of objective data is needed. For example, the information about the success of their own professional work (reports and opinions of mates  or patients, working history and so on). I also suggest to revise conclusiones and include reflection in relation to the goals of the study and new suggestions.

Author Response

Response to Reviewer 3 Comments
Thank you for your valuable contribution that improves this manuscript.

Point 1: Dear authors! The topic and the goals of the study are relevant and important, especially in conditions of world pandemic. I only have very little suggestions to improve your work. I think it is necessary to include more theoretic positions about the importance on the sense of humor in professional work. Also, it is important to present clearly each question or item of the test for the readers. I suggest to start the presentation of the data with general qualitative features of the responses to each item. I suggest to include concrete comments about the usefulness of each item or a kind of critical reflection about the items.  Probably, proposals about future studies or inclusion of new items or new design of the tests. It is also useful to think that psychological study may not be based only on self reports, but also a confirmation of objective data is needed. For example, the information about the success of their own professional work (reports and opinions of mates or patients, working history and so on). I also suggest to revise conclusiones and include reflection in relation to the goals of the study and new suggestions.

Response 1: We added further literature on the sense of humor in professional work and elaborated further the discussion on the basis of your comments. All the additions are highlighted in yellow in the resubmitted manuscript. The parts of specific interest are: lines 82-86, lines 96-101, lines 388-393 and lines 396-408.

Round 2

Reviewer 2 Report

I am satisfied with author(s) replies to my comments, thus, I endorse the publication in its current version.